# Medium-Scale UAVs: A Practical Control System Considering Aerodynamics Analysis

**Mohammad Sadeq Ale Isaac** [1,2,†] , **Marco Andrés Luna** [1,3,†] , **Ahmed Refaat Ragab** [3,4,5,†] , **Mohammad Mehdi Ale Eshagh Khoeini** [6,†] , **Rupal Kalra** [3,7,†] , **Pascual Campoy** [1,†] , **Pablo Flores Peña** [3,6,†] and **Martin Molina** [8,*,†]

1   Computer Vision and Aerial Robotics Group, Centre for Automation and Robotics (C.A.R.), Universidad Politécnica de Madrid (U.P.M.-CSIC), 28006 Madrid, Spain
2   Wake Engineering Company, 28906 Getafe, Spain
3   Drone-Hopper Company, 28919 Leganés, Spain
4   Department of Network, Faculty of Information Systems and Computer Science, October 6 University, Giza 12511, Egypt
5   Department of Electrical Engineering, University Carlos III of Madrid, 28919 Leganés, Spain
6   Department of Business Administration, Istanbul Aydin University, 34295 Istanbul, Turkey
7   Department of Aerospace Engineering, Universidad Politécnica de Madrid, 28040 Madrid, Spain
8   Department of Artificial Intelligence, Universidad Politécnica de Madrid, 28040 Madrid, Spain
\*   Correspondence: martin.molina@upm.es
†   These authors contributed equally to this work.

**Abstract:** Unmanned aerial vehicles (UAVs) have drawn significant attention from researchers over the last decade due to their wide range of possible uses. Carrying massive payloads concurrent with light UAVs has broadened the aeronautics context, which is feasible using powerful engines; however, it faces several practical control dilemmas. This paper introduces a medium-scale hexacopter, called the Fan Hopper, alimenting Electric Ducted Fan (EDF) engines to investigate the optimum control possibilities for a fully autonomous mission carrying a heavy payload, even of liquid materials, considering calculations of higher orders. Conducting proper aerodynamic simulations, the model is designed, developed, and tested through robotic Gazebo simulation software to ensure proper functionality. Correspondingly, an Ardupilot open source autopilot is employed and enhanced by a model reference adaptive controller (MRAC) for the attitude loop to stabilize the system in case of an EDF failure and adapt the system coefficients when the fluid payload is released. Obtained results reveal less than a 5% error in comparison to desired values. This research reveals that tuned EDFs function dramatically for large payloads; meanwhile, thermal engines could be substituted to maintain much more flight endurance.

**Keywords:** medium-scale UAV; adaptive control; motor failure; payload carriage

## 1. Introduction

Several research studies have been conducted to classify UAVs regarding their size, which finally, drive on micro, small, medium, and large scale platforms [1–4]. According to the mentioned categories, micro-scale and small-scale UAVs are those less than 25 kg, large-scale UAVs are more than 500 kg, and medium-scale UAVs are classified with less than 500 kg of gross takeoff weight, containing payloads up to 200 kg weight [1], which are currently spread in abundant applications, namely firefighting, irrigation, camera carriage, emergency and rescue, etc. Designing beneficial aerial systems requires high stability and safe flights. The former needs potent engines that work with tuned controllers, and the latter requires enclosed platforms and auxiliary actuator units to protect the structure and components and satisfy the standard control requirements. Moreover, to satisfy the mentioned applications, heavier payloads will aim to perform optimized missions at lower time stamps. Consequently, these drones necessitate quite a lot of power to carry out heavy

payload missions, and the EDFs generate this energy adequately, supplying electrical batteries but with lower autonomy than thermal resources. This comes from rechargeable batteries having a current useful energy density of around 120 Wh/kg, compared to 12,000 Wh/kg for fossil fuels; in other words, 1 kg of gasoline is equivalent to 25 to 30 kg of batteries [5]. In counterpart, due to thermal engines' drawbacks in emitting nitrogen oxide and carbon oxide ($NOx$ and $CO_2$) elements, mechanical uncertainties of many components required for propulsive systems, slower reactions, and wasting a significant volume of fuel, they are not highly efficient. Therefore, the dispute between the weight of the batteries and their sufficient energy leads to the use of electrical power sources.

On the other hand, powerful electrical engines that maintain higher thrusts for longer are practically targeted for heavy payloads, as mentioned in [6,7]. Performing the mentioned factors necessitates engines with the minimum energy waste that could be found in EDF types. Furthermore, another novel manner for low energy consumption is mentioned in [8], as the authors proposed dynamic energy saving for heavy drones in a swarm mission, concurrently optimizing the time and distance. Aerodynamically, tight ducted fans facilitate the actuators' function by reducing the blades' tip losses, which results in decrements in undesired yaw moments and produces more efficient thrust than convectional propellers of similar diameters [9–11]. Moreover, ducts protect the propellers, operate more efficiently at high airspeeds, reduce the blade tip speed and vorticity to be quieter, and produce higher static thrust than free propellers of the same size [12,13]. In contrast, ducted fans have instabilities, especially when leaving the near ground space [12], in transient mode between hover and forward flight, and when exposed to a wind gust, by reacting with a nose-up pitching moment [14]. The latter happens at a higher angle of attack (AoA), in which the duct produces aerodynamic drag [15]. Overall, the potential benefits outweigh the disadvantages, and the ducted fan geometry was pursued in this research, since concatenating six ducts symmetrically moves the center of mass (CoM) to the midpoint of the structure and eliminates static instabilities, as shown further in Figure 1. Several models are studied to overcome structural complexities, namely high vibration of the fuselage during flight and instabilities coming from the heavy payload, especially when it contains liquid, which produce extra forces and moments on the body system, where they might increase the system degrees of freedom (described further in Section 2). Through this research, a brief aerodynamic analysis is stated to show the weak and robust points of EDF design.

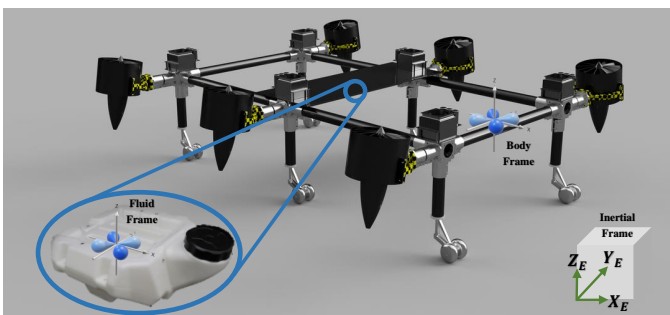

**Figure 1.** Demonstration of different coordinate systems, *Body* frame, *Fluid* frame, and the *Inertial* frame.

In comparison to traditional PID controllers, nonlinear techniques that conquer abrupt instabilities such as motor failure and payload release converge to stability faster, and among them, adaptive methods perform better tracking [16,17]. Accordingly, the authors of [18] presented an adaptive sliding mode controller (ASMC) augmented by a control allocation scheme to examine the benefits of the controller for a hexacopter platform, which resulted in better tracking against drastic faults through simulation in comparison to SMC; however, this work is limited to the simulation. Likewise, in [19], the authors concentrated on an adaptive neural PID controller, using an identification method for a discrete-time system to be developed in the real world. Furthermore, they compared the designed model

with static PID, and the one advanced with an extended Kalman filter (EKF) for better tracking adaptation. In addition, the authors of [20] studied a fuzzy adaptive fixed-time sliding mode controller (FAFSMC) on two and three-link manipulators as systems with high degrees of uncertainty. Then they affirmed the benefits of FAFSMC against non-fuzzy mode and non-fixed-time observers, concluding faster convergence in FAFSMC due to the fuzzy logic and the online estimators. Nonetheless, this work was also carried out on a simulation platform. The controller proposed in this paper is empowered by an adaptive reference model system, which proves asymptotic stability in a practical case utilizing a suitable Lyapunov candidate.

Additionally, considering a huge fuel tank as the payload requires complete mathematical modeling and then a controller law, leading the system to stability. During this research, a 35 kg hexacopter called Fan Hopper was modeled, designed, built, and flown. For the first design, a comprehensive aerodynamics analysis was conducted to determine the strength and weaknesses of the conceptual model and to investigate the controllability of such a model using turbofan engines. This paper is organized into nine sections, as follows; Section 2 describes medium-scale multicopters, the novel designed model, and aerodynamics considerations; Section 3 discusses a time-varying dynamic system of a multicopter; Section 4 describes the control algorithm implemented; Section 5 examines the results obtained from the simulation; Section 6 describes the hardware system and components utilized; Section 7 displays the results obtained for practical tests; Section 8 compares virtual and real results.

## 2. Medium-Scale Multicopters

Among medium-scale UAVs, multicopters have recently been more popular than other types because of their easy configuration, multifunctional usage, static stability, and vertical takeoff and landing (VTOL) [21,22]. Among multicopters, hexacopters offer more power, efficiency, and stability. Additionally, Hexa-kinds can handle the flight effectively enough to land safely in the case of failure and can carry greater payloads [21]. According to this research, a quadratic Hexa configuration is a better solution to manage heavy payloads and failure modes than a circular one, discussed further in Section 3. Further, the innovative configuration considered for this research is introduced.

### 2.1. Fan Hopper, a Novel Design for Medium-Scale Hexacopters

The multirotor under investigation through this research is called Fan Hopper; this platform has several complexities, from mechanical design to control algorithms, practical flight tests, and aerodynamic amendments. As shown in Figure 2, the drone has six EFDs installed inclined, varying the incident angle from 0° (vertical) to 22°, to maintain both thrust and cancel the yaw moment produced by each engine. This phenomenon occurs when all rotors rotate in the same direction, as in our investigation. Considering the geometry of EDFs, the yaw moment produced by each engine is minuscule but not zero; therefore, installing motors with an incident angle is an ingenious solution to control this small amount. The system is led by an open source autopilot (Pixhawk Cube), which allows development of the guidance loop (upper controller loop) using special plugins, which are described further in Section 4. Furthermore, the system contains a fluid payload tank under its CoG up to 20 kg weight that is dynamically modeled in Section 3.

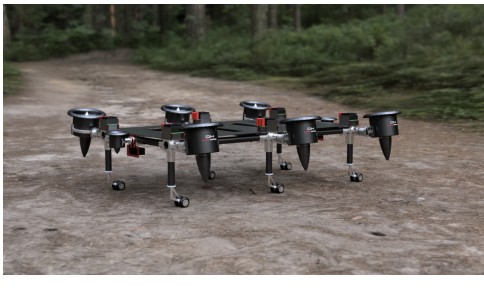 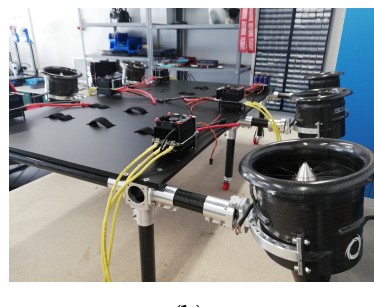

(**a**)　　　　　　　　　　　　　　(**b**)

**Figure 2.** A brief schematic of Fan Hopper's designed model; (**a**) the configuration with components installed as a whole; (**b**) the incident angle of the ducted fan.

### 2.2. Aerodynamics Considerations

Considering EDFs to lift the drone, several investigations were conducted, including an optimized aerodynamic configuration briefly approached as follows:

- Firstly, a single rotor with a 1 m diameter, 5 m/s inlet air velocity, and 300 rad/s rotor tangential velocity was considered. Obviously, due to the self-feeding toroidal vortex, asymmetry of the streamline aft and forward of the duct (triggered by the velocity of the rotor), high pressure down the rotor, disperse the droplets, as shown in Figure 3a, and the backflow area of the efficiency faced reduction; meanwhile, a huge stream rotation was observed, as shown in Figure 3b.

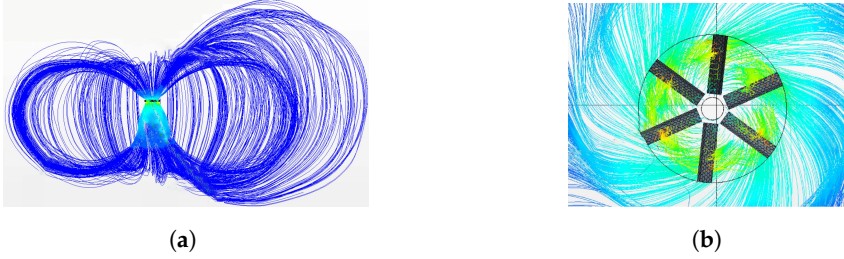

(**a**)　　　　　　　　　　　　　　(**b**)

**Figure 3.** Analysis of a single propeller; (**a**) asymmetry streamlines around the rotor; (**b**) the stream rotation.

- Secondly, the stator was added to eliminate the stream rotation, as shown in Figure 4a; this added a shrouded design which led the stream downwards and avoided vortex formation much better, as well as the backflow area; however, convergence of the stream was still complicated. As shown in Figure 4b, the droplet distribution was unrealistic due to lack of an actual injector. Additionally, the multiphase observed was adequate but complicated to match the transient approach.
- Thirdly, multiple ducts are concatenated to compensate for the instabilities in the transient mode and investigate the interaction between the ducts and the ground effects. Additionally, the propeller design was improved to give a realistic downstream, as shown in Figure 5a. Moreover, considering the airspeed as $v_a = 5$ m/s, the blade tangential tip speed as $\omega_t = 25$ rad/s, and the propeller diameter as $d_p = 1$ m, Equation (1) could be solved as follows:

$$TSR = \frac{\omega_t R}{v_a} = \frac{25\frac{rad}{s} * \frac{3.14}{2\pi}\frac{m}{rad}}{5\frac{m}{s}} = 2.5 \tag{1}$$

where TSR represents the tip speed ratio that equals 2.5, which is sufficient for a propeller of 5–6 blades. In addition, the total thrust diagram of different rotors was evaluated, which went up to 2700 N at the steady-state level; also, the medium mass flow reached 90 kg/s. Meanwhile, the absence of a multiphase model gave a better convergence, the downwash was realistic without the presence of backflow, and droplet ground impingement was corrected, as shown in Figure 5c. Furthermore, according to the table in Figure 5b, *rotor 12* and *rotor 22* had higher downforce than others, which overall stated the improved configuration of multiple rotors rather than a single one. The different rotor behaviors suggest a distinct injector strategy that even influences the UAV control.

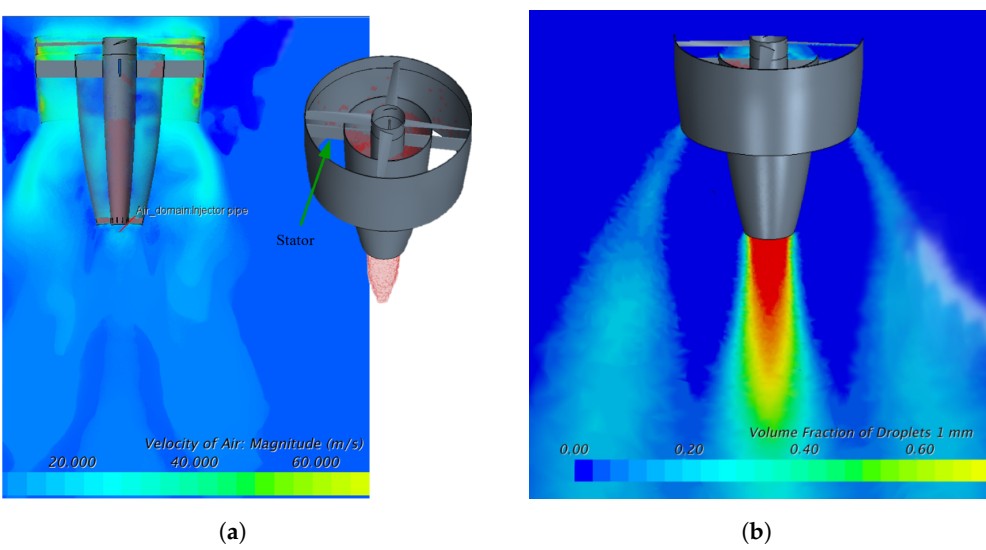

(**a**)          (**b**)

**Figure 4.** Analysis of a single propeller; (**a**) the rotor and shroud; (**b**) unrealistic droplet distribution due to no real injector.

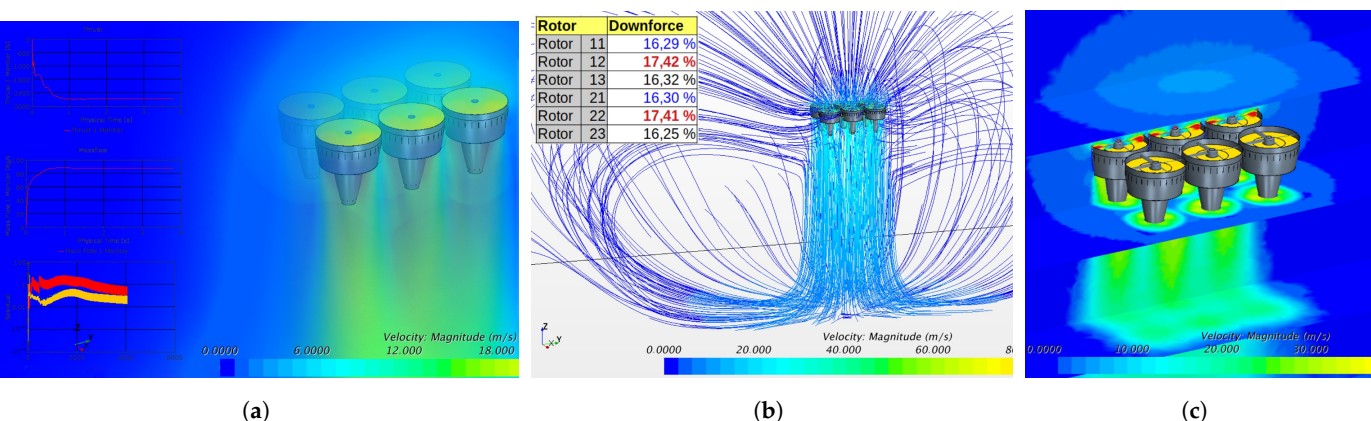

(**a**)          (**b**)          (**c**)

**Figure 5.** Analysisof 6 propeller engines; (**a**) (upper-left part) rotor thrust; (right part) mass flow of the stream passing through the engines; (**b**) streamlines around the model; (**c**) absence of multiphase model gave better convergence.

- Finally, regarding the volume fraction of vapors ($H_2O$) exhausted from the tube, as shown in Figure 6, numerous iterations were carried out, multiple injector models were developed, excellent results were obtained due to several multiphase calculations, different droplet sizes were evaluated, and the ground impregnation was improved.

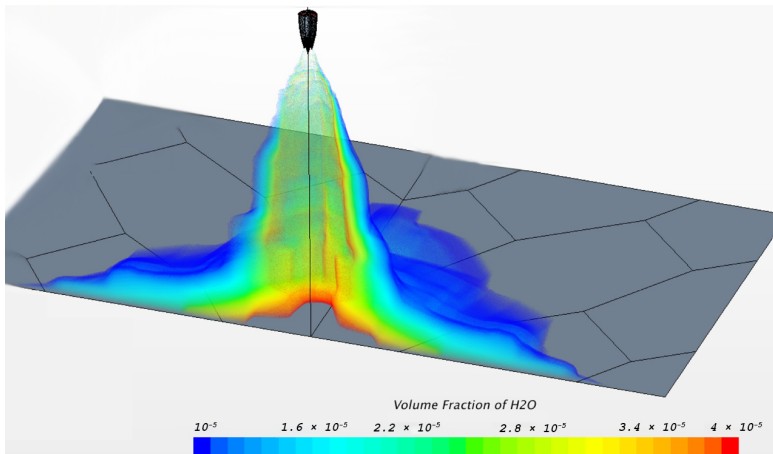

**Figure 6.** Injectorsdeployed to make the droplet distribution realistic; the color distribution relates to the $H_2O$ volume fraction.

### 3. Hexacopter Time-Varying Dynamics

Regardless of general dynamics, substantial equations of conventional multirotors have been described several times through research works such as [23–25]; considering a fluid payload under the center of mass (CoM) of the UAV, the moving fluid generates a considerable sloshing effect [26]; depending on the repository shape, baffle balls, and baffle walls, for the same amount of fluid, the displacement of the center of gravity (CoG) of the fluid causes less slosh height with a baffle wall and even less in the presence of a baffled ball inside the tank. Meanwhile, among several types of tank shapes, hexagonal geometry functions more appropriately than circular or rectangular geometry. Therefore, in this research, a hexagonal box is utilized with a baffled ball inside using an optimized configuration.

Furthermore, to determine an acceptable solution for the sloshing problem, assuming Newtonian viscous incompressible flow and constant density, the results of a computational investigation of a similar case are employed [27]. Regardless of the fluid movements during the flight, the fluid mass matrix is constant, which leads the centrifugal and Coriolis forces to zero, so that for an element, this could be rewritten as in Equation (2).

$$M_e = \int_{V_e} \rho_e S_e^T S_e \, \mathrm{d}V_e \tag{2}$$

where, $\rho_e$ and $V_e$ are the density and volume of the fluid element, $S_e$ is the matrix of element shape function, and $*^T$ states the transposition of that matrix. Then, the total force of the fluid could be proposed as a summation of its inertia force, body force, inertial elastic forces (viscous force and stress force), surface traction force, and penalty force due to the incompressibility assumption, as elaborated in [27–29], and summarized in Equation (3).

$$F_T = \int_{V_e} F_{l,e}^T S_e \delta e_e \, \mathrm{d}V_e + \int_{A_e} N_e^T \mu_{e1} S_e \, \mathrm{d}A_e - \int_{V_e} \mu \delta J_e J_e^{-1} \, \mathrm{d}V_e - \int_{V_e} (\rho_e a)^T S_e \delta e_e \, \mathrm{d}V_e \tag{3}$$

where $F_l$ is the body force vector of the fluid element, $\delta e_e$ is the differential vector of the element nodal coordinate, $N_e$ is the vector normal to the surface of the element, $\mu_e 1$ is the first stress tensor of Piola–Kirchhoff, $\mu$ is Cauchy symmetry stress tensor, $a$ is the acceleration vector of the fluid element, $J_e$ is the matrix of position gradients vector of the fluid element, $J_e = \begin{pmatrix} r_x & r_y & r_z \end{pmatrix}$, and the determinant of this matrix equals one, and $a$ is the acceleration vector of the fluid element.

Equations (2) and (3) are according to the final element (FE) absolute nodal coordinate formulation (ANCF), considering the continuity of the fluid element interface inside the tank and based on the Eulerian approach. Moreover, the nonlinearities of the tank fluid were negligible because of the lower velocity domain of a medium-scale multicopter; for high velocities and maneuvers causing severe movements, however, the fluid elements

could be described as eight nodal bricks, each one interpreted into a polynomial function of 32 coefficients, and consequently, the fluid element system could have 96DoF, as mentioned in [27], which should be linearized for a whole dynamical system. Considering the time-varying inertia effect of the fluid tank, the translational motion equation of the drone as a whole is driven by Equation (4). All the equations relate to the body coordinate, following the north-east-down (NED) coordinates with reversed *Z axis*, as shown in Figure 1. Then, to reduce the negativity of coefficients, *Z axis* of the *Body* frame was chosen to be reversed regarding the NED system.

$$\dot{r}_e = R_{eb}^T V_b + w_e \times r_e \tag{4}$$

where $\dot{r}_e$ denotes the CoM position vector derivative in the *Inertial* frame, $R_{eb}$ is the rotation matrix from *Inertial* to frame *Body* frame, described in [25,27], $V_b$ is the CoM relative velocity in *Body* frame, and $w_e \times r_e$ is the air absolute velocity at $r_e$. Then, the translational dynamics of the system are as follows in Equation (5), considering the Coriolis formulation of $\frac{d}{dt}V_b = \dot{V}_b + \omega_b \times V_b$.

$$F_b = \dot{V}_b m_f + \dot{m}_f V_b + R(\dot{m}_f - \omega_e)(w_e \times r_e) - (\omega_b + R\omega_e) \times V_b + Rg \tag{5}$$

where $F_b$ is the total force impacting on *Body* frame, $m_f$ is the fluid tank mass, $\dot{m}_f$ is the fluid-derived mass that is negligible for solid parts and considerable for the fuel part, $\omega_e$ and $\omega_b$ are the drone angular velocity in *Inertial* and *Body* frames, respectively, and $g$ is the gravitational acceleration.

## 4. The Proposed Control Strategy

Considering the dynamics equations described through Section 2, a nonlinear controller is proposed to maintain the stability of the drone against the engine's failure. Among nonlinear methods, model reference adaptive control (MRAC) is a fine solution in the case of releasing payloads such as the Fan Hopper, since the total mass as an internal uncertainty is sustainably changing and the controller must adapt the parameters hastily. Then, to perform an agile reaction, the stability is established by a Lyapunov function to converge the model to the domain of attraction and stabilize the system. As shown in Figure 7, a brief schematic diagram of the adaptive controller is demonstrated regarding the inner loop. Differing from the reference and proposed model systems, an adaptive rule is defined to enter the state space, compute the position and velocity errors, and update the matrices as any internal error occurs. Multiplying the update matrix to the state model results in updated inputs for engines to be applied, then differing them to the errors obtained from the state space model concludes in EDF commands. Considering the state input vector of the system as *X*, it yields:

$$X = \begin{bmatrix} x & \dot{x} & y & \dot{y} & z & \dot{z} & \phi & \dot{\phi} & \theta & \dot{\theta} & \psi & \dot{\psi} \end{bmatrix}^T \tag{6}$$

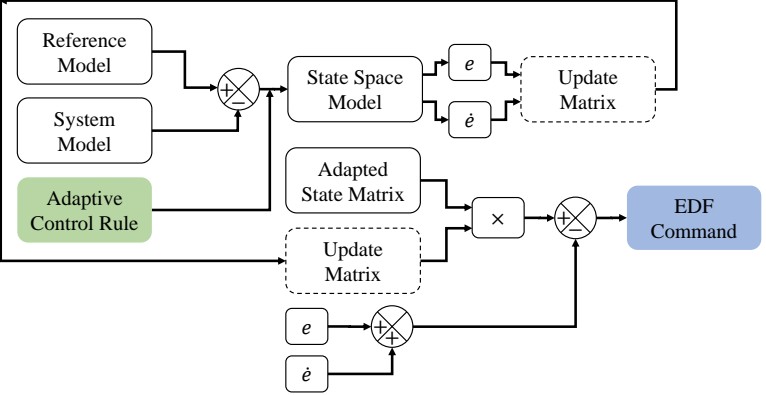

**Figure 7.** Diagram of implemented adaptive controller for attitude loop.

where the goal is to approximate the parameters of a model adapted to the main system, we express the reference model as a nonlinear second-order robot in state space as $G_m(s)$:

$$G_m(s) = \frac{w^2}{s^2 + 2\xi\omega s + \omega^2} \rightarrow\rightarrow \ddot{\chi}_m + \underbrace{2\xi\omega}_{m_1}\dot{\chi}_m + \underbrace{\omega^2}_{m_2}\chi_m = \underbrace{\omega^2}_{n} u_c \tag{7}$$

where $m_1$, $m_2$, and, $n$ are reference parameters, and $u_c$ is the general controller input. Assuming $G(s)$ is a proposed second-order model to reduce the difference with the reference model, it gives

$$G(s) = \frac{1}{s(s+m)} \rightarrow \ddot{\chi} + m\dot{\chi} = u \tag{8}$$

where $m$ is the parameter to be estimated when the adaptive model approaches the reference model. We determine the control rule for the second order system as follows:

$$u = \alpha_1 u_c - \alpha_2 \dot{\chi} - \alpha_3 \chi \tag{9}$$

Taking the mass uncertainty into account, it differs as the payload tank releases the fluid, which is computed by update values $\alpha_i, i = \{1, 2, 3\}$ for the adaptation law. Differentiating Equations (7) and (8)

$$\ddot{\chi} - \ddot{\chi}_m - m_1\dot{\chi}_m - m_2\chi_m = -(m + \alpha_2)\dot{\chi} - \alpha_3\chi + (\alpha_1 - n)u_c \tag{10}$$

Considering $e = \chi - \chi_m$, $\dot{e} = \dot{\chi} - \dot{\chi}_m$, $\ddot{e} = \ddot{\chi} - \ddot{\chi}_m$, and adding $m_1\dot{\chi} + m_2\chi$ to both sides of the equation,

$$\ddot{e} + m_1\dot{e} + m_2 e = -\underbrace{(m + \alpha_2 - m_1)}_{\beta_2}\dot{\chi} - \underbrace{(\alpha_3 - m_2)}_{\beta_3}\chi + \underbrace{(\alpha_1 - n)}_{\beta_1} u_c \tag{11}$$

where $e$ denotes the error between the reference and adaptive values. Hence:

$$e = \frac{1}{s^2 + 2\xi\omega s + \omega^2}[\dot{\chi} \quad -\chi \quad u_c][\beta_2 \quad \beta_3 \quad \beta_1]^T \tag{12}$$

Obviously in Equation (12), the numerator and denominator difference degree is more than one; therefore, it is not strictly positive real (SPR), and Kalman Yakubovich's Lemma [30] is not usable, while it is solvable utilizing state space equations, as follows:

$$\begin{cases} \dot{X} = AX + BU \\ Y = CX + DU \end{cases} \tag{13}$$

$$A = \begin{bmatrix} O_{6\times6} & I_{6\times6} - a_2 \\ O_{6\times6} & -a_1 \end{bmatrix} = \begin{bmatrix} 0\,0\,0\,0\,0\,0 & 1-\omega_\phi^2 & 0 & 0 & 0 & 0 & 0 \\ 0\,0\,0\,0\,0\,0 & 0 & 1-\omega_\theta^2 & 0 & 0 & 0 & 0 \\ 0\,0\,0\,0\,0\,0 & 0 & 0 & 1-\omega_\psi^2 & 0 & 0 & 0 \\ 0\,0\,0\,0\,0\,0 & 0 & 0 & 0 & 1-\omega_\phi^2 & 0 & 0 \\ 0\,0\,0\,0\,0\,0 & 0 & 0 & 0 & 0 & 1-\omega_\theta^2 & 0 \\ 0\,0\,0\,0\,0\,0 & 0 & 0 & 0 & 0 & 0 & 1-\omega_\psi^2 \\ 0\,0\,0\,0\,0\,0 & -2\xi_\phi\omega_\phi & 0 & 0 & 0 & 0 & 0 \\ 0\,0\,0\,0\,0\,0 & 0 & -2\xi_\theta\omega_\theta & 0 & 0 & 0 & 0 \\ 0\,0\,0\,0\,0\,0 & 0 & 0 & -2\xi_\psi\omega_\psi & 0 & 0 & 0 \\ 0\,0\,0\,0\,0\,0 & 0 & 0 & 0 & -2\xi_\phi\omega_\phi & 0 & 0 \\ 0\,0\,0\,0\,0\,0 & 0 & 0 & 0 & 0 & -2\xi_\theta\omega_\theta & 0 \\ 0\,0\,0\,0\,0\,0 & 0 & 0 & 0 & 0 & 0 & -2\xi_\psi\omega_\psi \end{bmatrix} \tag{14}$$

$$B = \begin{bmatrix} 0\,0\,0\,0\,0\,1/m & 0 & 0 & 0 & 0 & 0 \\ 0\,0\,0\,0\,0\,0 & l/I_x & 0 & 0 & 0 & 0 \\ 0\,0\,0\,0\,0\,0 & 0 & 0\,l/I_y & 0 & 0 \\ 0\,0\,0\,0\,0\,0 & 0 & 0 & 0 & 0\,1/I_z \end{bmatrix}^T, C = [0 \cdots_4 0\,1 \cdots_4 1]_{1\times12}, D = O_{12\times4} \tag{15}$$

where, $A \in \mathbb{R}^{12 \times 12}$ denotes the state coefficients matrix generalized of Jacobian matrix, $B \in \mathbb{R}^{12 \times 4}$ expresses the controller inputs to state matrix, $U \in \mathbb{R}^4$ is the controller inputs vector, $C \in \mathbb{R}^{1 \times i}$ is the state-to-output matrix depending on $j$ demands, which are $\phi, \theta, \psi, \dot{\phi}, \dot{\theta}, \dot{\psi}$ for this research, $D \in \mathbb{R}^{12 \times 4}$ is the static gain matrix, which represents the output-to-input ratio, $\xi_i$ and $\omega_i, i = \{\phi, \theta, \psi, \dot{\phi}, \dot{\theta}, \dot{\psi}\}$ are calculated iterating the simulation numerous times. Additionally, the parameters in the $B$ matrix are linearized fast dynamics of the hexacopter, coming from $\Delta U_1 = m\ddot{z}$, $U_2 = I_x\ddot{\phi}$, $U_3 = I_y\ddot{\theta}$, $U_4 = I_z\ddot{\psi}$, in which the $m$ refers to the whole drone mass, containing $m_f$ mentioned in Equation (5). Determining the errors in state space, it gives

$$\begin{bmatrix} \dot{e} \\ \ddot{e} \end{bmatrix} = A \begin{bmatrix} e \\ \dot{e} \end{bmatrix} + B[-\dot{\chi} \quad -\chi \quad u_c][\beta_2 \quad \beta_3 \quad \beta_1]^T \tag{16}$$

To obtain the update matrix for attitude values and rates, $F \in \mathbb{R}^{6 \times 6}$ and $P \in \mathbb{R}^{6 \times 6}$ are defined as diagonal positive definite (DPD) matrices. The former contains $f_i, i = \{f_\phi, f_\theta, f_\psi, f_{\dot{\phi}}, f_{\dot{\theta}}, f_{\dot{\psi}}\}$ as diagonal elements equal to eigenvalues, and the latter satisfies the condition $A^T P + PA = -I$, in which $I$ is the identity $6 \times 6$ matrix. Thus, solving Equation (16) and the latter criterion [31,32], the update matrix yields:

$$[\dot{\beta}_2 \quad \dot{\beta}_3 \quad \dot{\beta}_1]^T = -F[-\dot{\chi} \quad -\chi \quad u_c]^T B^T P \begin{bmatrix} \dot{e} \\ \ddot{e} \end{bmatrix} \tag{17}$$

On the other hand, in terms of stability, a positive definite function (PDF) is proposed as a Lyapunov candidate, and its derivative must be demonstrated to be negative definite.

$$V = \frac{1}{2} e^T P e + \beta^T F \beta \tag{18}$$

where $V$ denotes the Lyapunov function, $e$ is the aforementioned error term of $\mathbb{R}^6$ (refers to Equation (11)), and $\beta$ is the update matrix defined previously. Meanwhile, $F$ appeared as a matrix and functions as a dot product due to being diagonal between two vectors; consequently, both terms result in $[\cdots]_{1 \times 6}[\cdots]_{6 \times 6}[\cdots]_{6 \times 1} = N_{1 \times 1}$ dimension. Then, the derivative of the Lyapunov candidate simplifies as follows:

$$\begin{aligned}
\dot{V} &= \frac{1}{2}(\dot{e}^T P e + e^T P \dot{e}) + \dot{\beta}^T F \beta + \beta^T F \dot{\beta} \\
\rightarrow \dot{V} &= \frac{1}{2}((\dot{x} - \dot{x}_m)^T P e + e^T P(\dot{x} - \dot{x}_m)) + 2F\beta^T \dot{\beta} \\
\rightarrow \dot{V} &= \frac{1}{2}((A_m e + \underbrace{\cdots}_{\zeta\beta})^T P e + e^T P(A_m e + \zeta\beta)) + 2F\beta^T \dot{\beta} \\
\rightarrow \dot{V} &= \frac{1}{2}e^T \underbrace{(A_m^T P + pA_m)}_{-I,\ (19)} e + \underbrace{\beta^T \zeta^T P e}_{\Delta} + \underbrace{e^T P \zeta\beta}_{\Delta^T} + 2F\beta^T \dot{\beta} \\
\rightarrow \dot{V} &= \underbrace{-\frac{1}{2}e^T I e}_{ND} + \frac{1}{2}\beta^T \underbrace{(Pe + e^T P \zeta\beta + 4F\beta^T \dot{\beta})}_{\text{to be equal } 0}
\end{aligned} \tag{19}$$

where $A_m$ is the Jacobian matrix of the reference model, and $\zeta$ is a synoptic notation of update matrix products. The first term of the simplified derivative function is ND; regarding the second term, the assumption is that $\beta^T = 0$ is not acceptable, and only $Pe + e^T P \zeta\beta + 4F\beta^T \dot{\beta}$ must equal zero to satisfy the definite negativeness of the term.

## 5. Simulation Results

Through this research, a complete virtual model of the Fan Hopper was designed to apply the controller algorithm to a 3D model and prevent multiple failures in the real world. Meanwhile, several random noises were imported, including the fluid release and a motor failure that examined the controller's functionality to update adaptive parameters. The simulation was carried out employing the open dynamic engine (ODE) of Gazebo. Thanks to the asset loading and unloading feature, Gazebo makes the simulation drive realistic. Concurrently, it can connect to the robotic operating system (ROS), which handles the movements, orientation, various sensors, and the control algorithm. Preparing for the simulation, all the cad models (refer to Section 6.1) were exported to *\*.urdf* format, using the ROS-URDF library (accessed on 3 July 2022), in order to prepare the mathematical for the simulation, as shown in Algorithm 1.

---

**Algorithm 1** FanHopper URDF Configuration.

---

**Require:** geometric params.
**Ensure:** the mathematical model
  $< robot\ name\ space =" FanHopper" >$
    $< linkname =" \textbf{fuselage}" >$
      $< inertial > \cdots < origin >, < mass >, and\ < inertia > \cdots < /inertial >$
      $< visual > \cdots < origin >, < geometry >, and\ < material > \cdots < /visual >$
      $< collision > \cdots < origin > and < material > \cdots < /collision >$
    $< /link >$
    $< \textbf{propellers}\ joint >$
      $< origin >< parent >, < child >, < axis >, < dynamic >, and\ < limit >$
    $< /joint >$                        ▷ six propellers
    $< gazebo\ reference =" \textbf{sensors}"\ joint >$
      $< type > and < plugin$
    $< /gazebo >$        ▷ IMU, forces, moments, and camera controllers
  $< /robot >$

---

As described in Algorithm 1, two hierarchies were applied; the *base link*, which is mentioned as *fuselage*, and *propellers*, which are dependent joints to the root. Then, all sensors used to communicate with ROS are at the same level defined as *reference*. Connecting the simulation with an adaptive algorithm elaborated in *python*, sensor data were subscribed to, and control regulations were applied by the ROS publishers. Several results were obtained by enhancing the controller algorithm, which briefly includes engine failure during the flight. As shown in Figure 8b, a semi-sine wave trajectory was simulated to examine the adaptive controller during a 5 min period. Accordingly, the purple lines highlight the control inputs and the blue lines correspond to the actual behavior of the system. Upon starting the mission, the drone lifted off to a 2.5 m altitude and followed the horizontal trajectory with less than 1% error, as shown in Figure 8a,d,g. Though after 2.7 min, one of the EDFs failed, the drone maintained stability, as shown in Figure 8b,c,e,f, and the only visible error appeared in the yaw angle, as shown in Figure 8h,i, which is even about 5° that could be neglected.

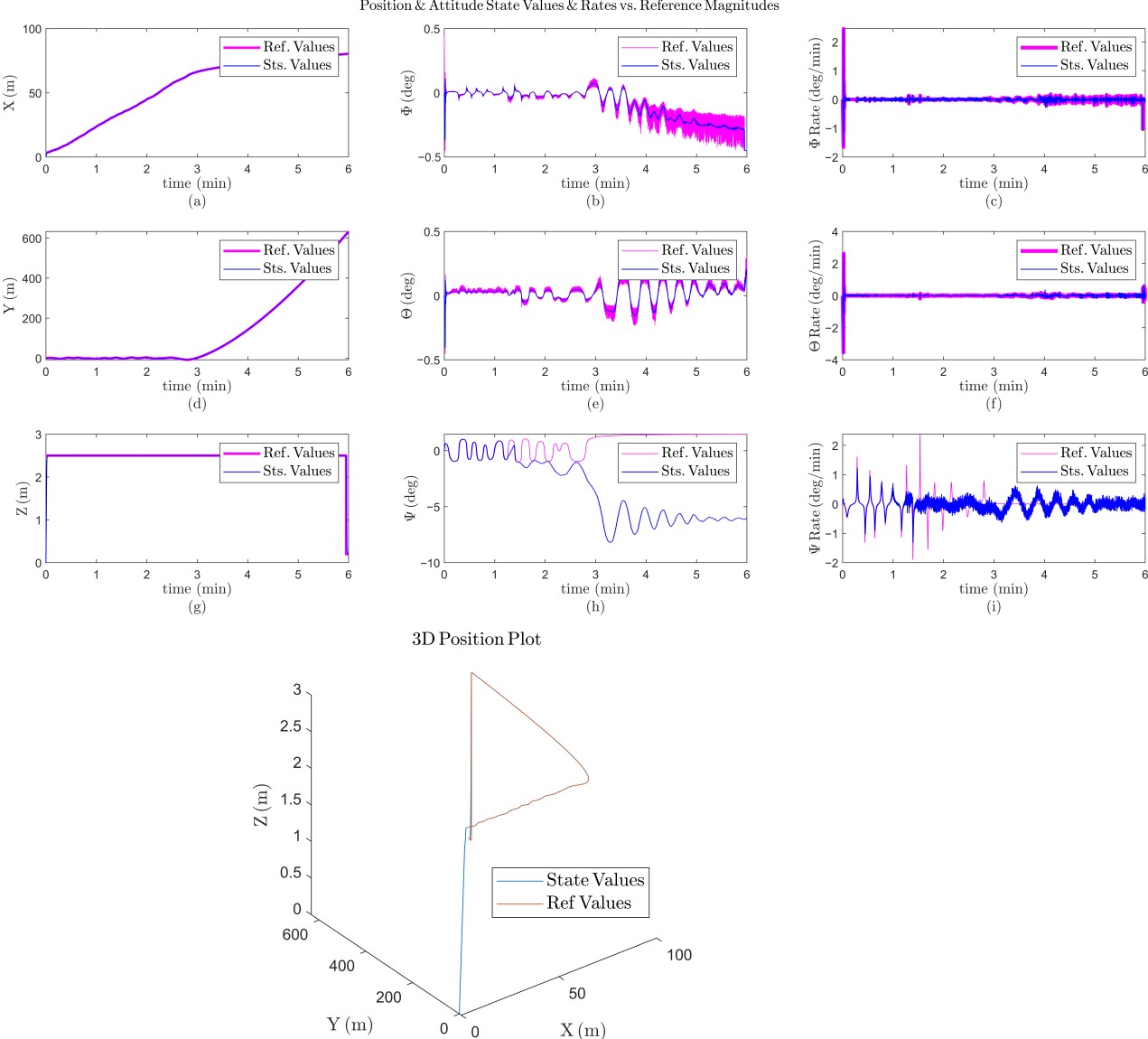

**Figure 8.** Simulation results when an EDF fails during 5 min; (**a**) the horizontal (*x*) position variation versus time; (**b**) the roll (*ϕ*) angle variation versus time; (**c**) the roll rate ($\dot{\phi}$) variation versus time; (**d**) the horizontal (*y*) position variation versus time; (**e**) the pitch (*θ*) angle variation versus time; (**f**) the pitch rate ($\dot{\theta}$) variation versus time; (**g**) the vertical (*z*) position variation versus time; (**h**) the yaw (*ψ*) angle variation versus time; (**i**) the yaw rate ($\dot{\psi}$) variation versus time.

## 6. Hardware Design

The design process followed in this research consists of three main steps: preliminary model design, assembly of the components, and mechanical and electrical integration. Then, multiple components are reviewed, redesigned, and reconstructed to guarantee a safe flight. Among possibilities for a hexacopter frame, a rectangular platform was chosen since a heavy payload shall be carried; when this was added to the electrical power system, which exceeded five kg in weight, it made the configuration rectangular. Additionally, as described in the Section 2.1, ducts that generated yaw moments were canceled out thanks to the quadratic frame.

### 6.1. Model Design and Assembly

Briefly, each component was designed in Microsoft Fusion, as shown in Figure 9. The drone fuselage has 97 cm width, 40 cm height with open wheels, is and 1090 cm

long, containing inclined EDFs installed, as shown in Figure 9a,b,d. The fluid tank was installed under the CoM of the drone. For instance, several component models are drawn in Figure 9e–g, including the novel designed connectors function to adjust the install angle of EDF that functions as a thrust vector, configurable between 0° and 22°. In addition, Figure 10 demonstrates the differentiation of the thrust and yawing torque of a Fan Hopper's EDF, respectively, and in a period of 30 s, based on the variation of the installment angle from 0° and 22°. Observing the diagram, the most optimized line belongs to the one with more thrust and less torque to cancel. Furthermore, the highest and lowest thrust happened at 15° and 20°, respectively; however, the absolute torque magnitude increased alongside the angle. Additionally, regarding the effective thrust timestamp during seconds 16 and 23, the thrust maximum difference amount was ≤100 N, and for torque, it was ≤0.05 N.m. The former amount is considerable and weighted the latter, since the torque was canceled out further by other EDFs in the Hexa configuration. Hence, considering detailed values, the optimized incident angle domain was chosen between 8° and 11° to perform the practical tests. In addition, Figure 10b displays the EDF vibration increasing as the installment angle raises, which is exempted near to 15°, which was related to a higher thrust. Therefore, this diagram again confirms the chosen domain in terms of vibrations.

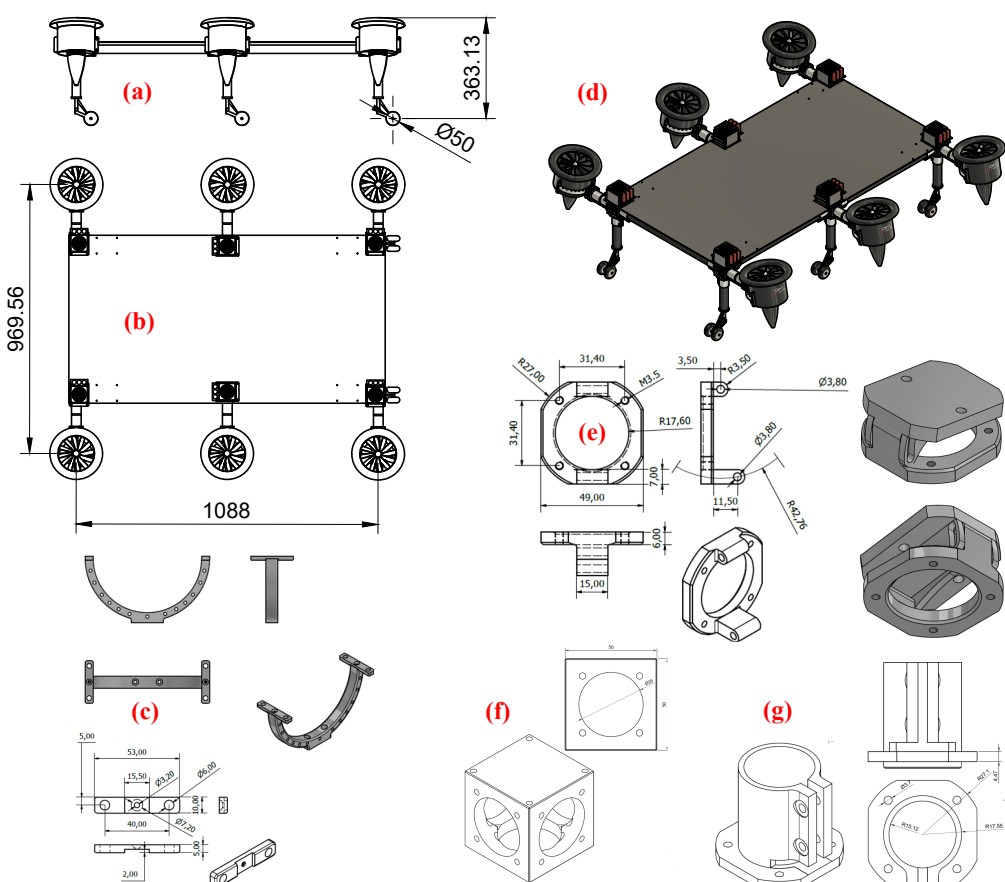

**Figure 9.** The CAD models of the Fan Hopper; (**a**) side elevation of the CAD model; (**b**) top view of the CAD model; (**c**) engine arm connectors; (**d**) a 3D view of the CAD model; (**e**) incident angle adjusters for duct engines; (**f**) the conjunction main connector; (**g**) body to duct arm connectors.

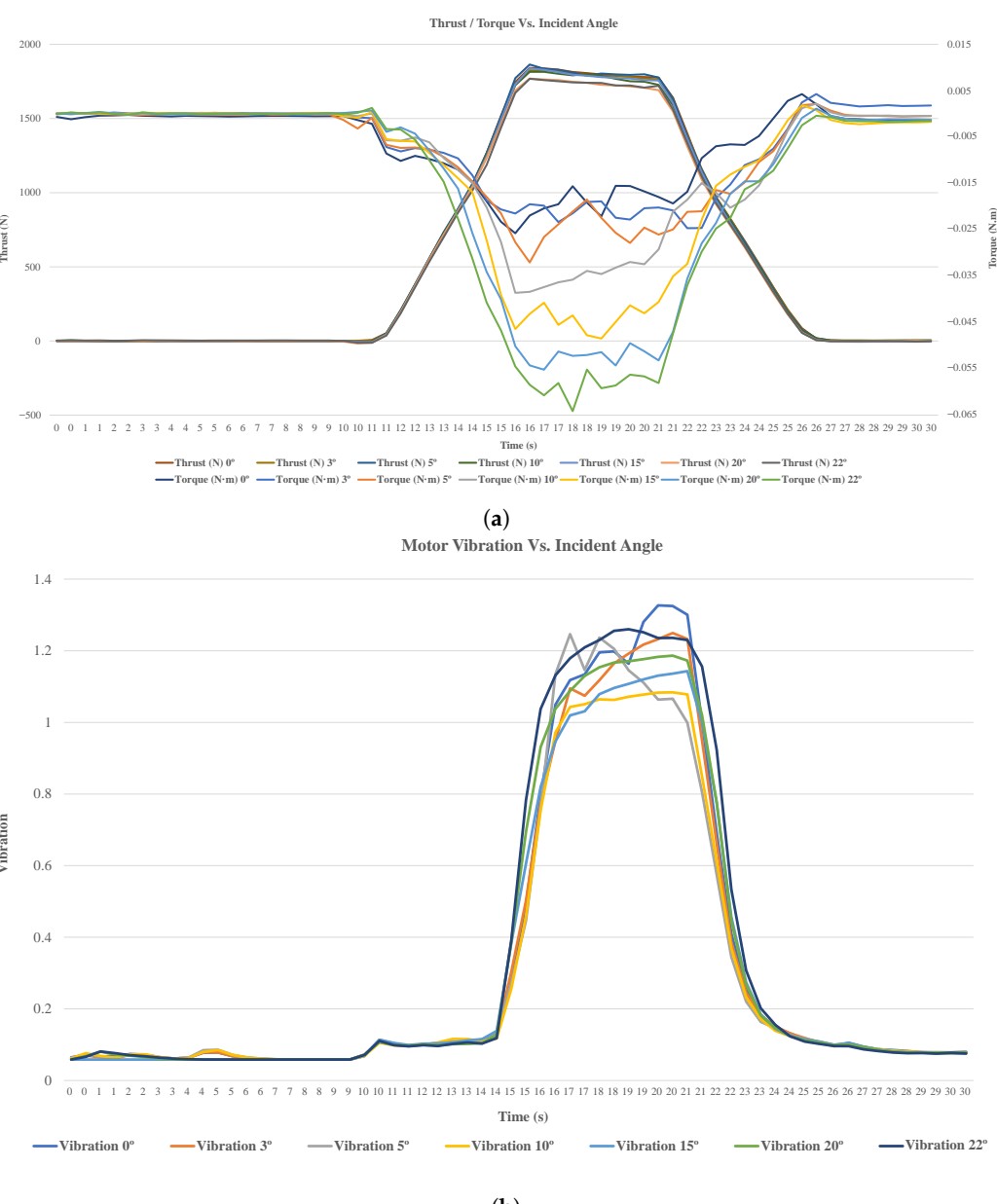

**Figure 10.** Diagram of an EDF data versus incident angles (0°–22°) during 30 s; (**a**) diagram of EDF thrust and yawing torque, (**left bar**) thrust values, (**right bar**) torque values; (**b**) the expressed vibration impact of distinct incident angles on the duct.

### 6.2. System Integration

Following the assembly, concatenated components as a whole are shown in Figure 9d,b, and the corresponding actual model is displayed in Figure 11f. Thanks to adjustable joints and landing bases, EDFs could be installed closer to the base link, and the landing basis varies according to the fluid tank dimension, as elaborated in Figure 12. Additionally, the initial characteristics of the Fan Hopper are stated in Table 1. Based on the aerodynamic analysis reviewed in Section 2.2, a suitable EDF configuration was selected for the Fan Hopper to facilitate a large payload; each SCHUBELER HST engine lifts to 10 kg weight and enhances the stator aerodynamics, with its efficiency maximized to 70%.

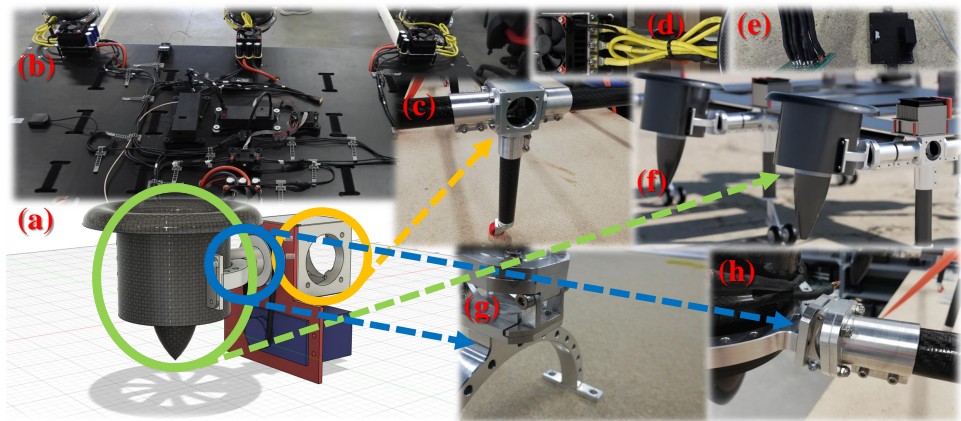

**Figure 11.** Assembled model of the Fan Hopper; (**a**) the EDF model connected to the joints, modifiable by the rubber band; (**b**) the base link, containing AP, connectors, antenna, fan, joints, wires, etc.; (**c**) the aluminum joint for the landing gear and motor arm; (**d**) the ESC cooler; (**e**) power distribution board; (**f**) the EDF system; (**g**) the duct holder; (**h**) the configurable duct joint to the arm.

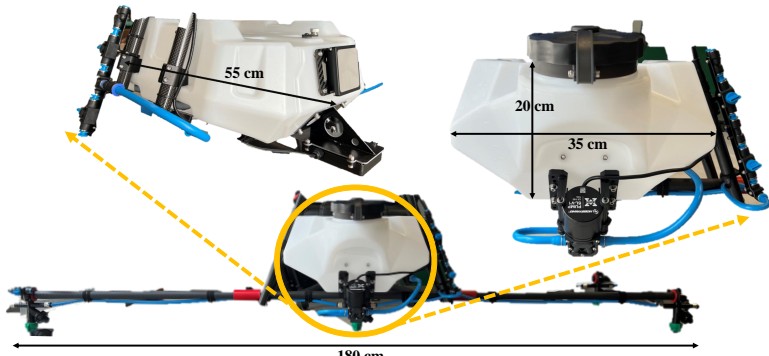

**Figure 12.** the integrated fluid tank schematic of the Fan Hopper; electrical components, valves, and tubes could be seen precisely on the left and right sides.

**Table 1.** Initial characteristics of the of the Fan Hopper.

| Parameter | Description | Weight (kg) |
|:---:|:---:|:---:|
| AP | Pixhawk Standard Set Cube Orange + ADS-B | 0.32 |
| EDF | SCHUBELER, *DS-98-DIA* HST | $1330 \times 6$ |
| GCS | modified QGroundControl, using *Qt*15.5.2 | - |
| ESC & Fan | | $0.52 \times 6$ |
| Battery | Quantum 5000 mAh | $2.215 \times 6$ |
| Base link | Wood and flexible materials | 5 |
| Cabling | - | 1 |
| Fluid tank | Containing the spray system and the regulator | 3 |
| | TOTAL empty weight | 35 |

Regarding the Section 2, the motor mixing configuration is described in Table 2. The numeration direction of the EDFs is counter clockwise, and central EDFs (2, 5) do not collaborate for pitch and yaw moments.

**Table 2.** Motor mixing configuration of the Fan Hopper.

| EDF | Roll Moment | Pitch Moment | Yaw Moment | Configuration |
|-----|-------------|--------------|------------|---------------|
| R1  | 0.5         | −1           | 0.5        |               |
| R2  | 0.5         | 0            | 0          | 1 ↻ —— ↻ 6    |
| R3  | 0.5         | 1            | −0.5       | 2 ↻ —— ↻ 5    |
| R4  | −0.5        | 1            | 0.5        | 3 ↻ —— ↻ 4    |
| R5  | −0.5        | 0            | 0          |               |
| R6  | −0.5        | −1           | −0.5       |               |

## 7. Practical Results

Several analyses were conducted to ensure the components' functionality and safety issues, including EDF, ground and balance, and flight tests. As shown in Figure 13, multiple balance tests were performed, once with two payload packages, each one weighing 5 kg, and then with four packages in symmetric order. Moreover, the total Fan Hopper weight is 35.4 kg, and the wooden ballast stabilizer below the drone has a 4 kg weight. During the first flight, numerous vibrations were observed in the base link, which decreased significantly using the adaptive method described in Section 4 by tuning the update matrix parameters. Meanwhile, several modifications were applied to the base link to make it more flexible against undesired vibrations. Previously, a complete practice was performed using the default parameters of the AP (Pixhawk) to affirm MRAC estimated values. In addition, results of two flight tests with altitudes higher than 5 m are shown, once with random low noises and then, with higher noises applied by the pilot.

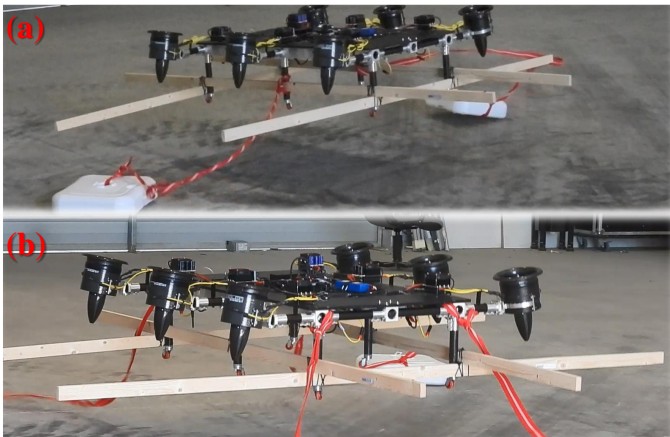

**Figure 13.** Ground tests made to assure the stability of Fan Hopper, and examine the lifting power; (**a**) lifting two payloads of 5 kg; (**b**) lifting four payloads of 5 kg.

The analytical results of a balance test are shown in Figure 14. This test was conducted near the ground with varying altitudes up to 1.5 m with different payload weights. Moreover, the flight was carried out with a copilot, i.e., piloting manually saturated by AP limitations to test its reactions. Meanwhile, the payload was separated by a rope further from the fuselage to prevent any accidents. During the test, in the time interval between 6 s to 7.5 s, the pilot applied 15° roll and pitch commands simultaneously with a random yaw command, which is demonstrated in Figure 14a–c. Although there was an alteration in attitude angles, the controller compensated the stability rapidly in less than 2 s, with minor errors.

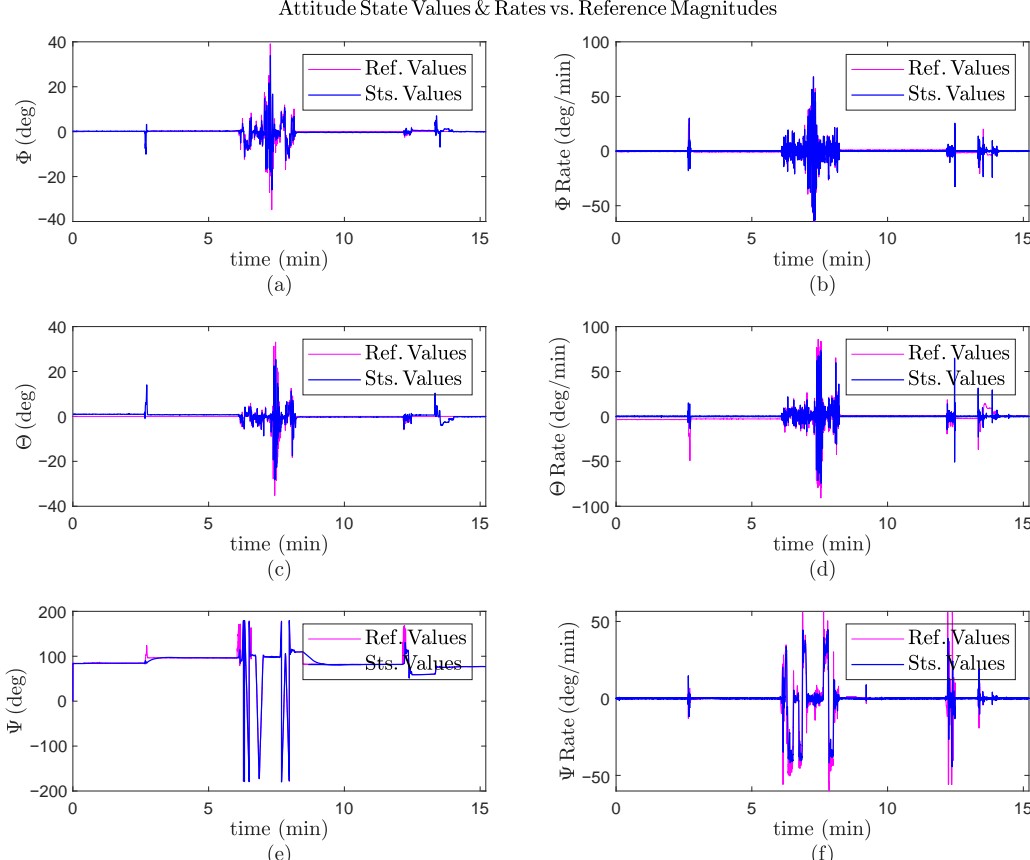

**Figure 14.** Balance test results near the ground during 15 min; (**a**) the roll ($\phi$) angle variation versus time; (**b**) the pitch ($\theta$) angle variation versus time; (**c**) the yaw ($\psi$) angle variation versus time; (**d**) the roll rate ($\dot{\phi}$) variation versus time; (**e**) the pitch rate ($\dot{\theta}$) variation versus time; (**f**) the yaw rate ($\dot{\psi}$) variation versus time.

Multiple flight tests at higher altitudes were conducted after configuring the ground tests and tuning the PID parameters to ensure the controllability of the FanHopper. The first results relate to a flight test during 18 min, containing a sudden pilot disturbance after passing 7.5 min, in which the attitude controller had an undershoot reaction due to saturating inputs, as shown in Figure 15a. The stability was regained in less than 10 s, since the deviation was canceled by a cascade approach, eliminating errors by regulating both the roll angle and its rate, as shown in Figure 15b. Likewise, unexpected phenomena were applied to the pitch and yaw angles that were damped rapidly after 7.5 min, 9.5 min, and 13 min.

Finally, the last results contain high disturbances leading to instability; however, they were damped with adequate haste. As shown in Figure 16, after 10 min, all attitude angles were altered simultaneously, as 10° for roll, 8° for pitch, and up to −200° for yaw, but were regulated in less than 2 s. At 15 min, one of the EDFs was unplugged from the system, which triggered a negligible transient error (less than 1°) to the pitch angle.

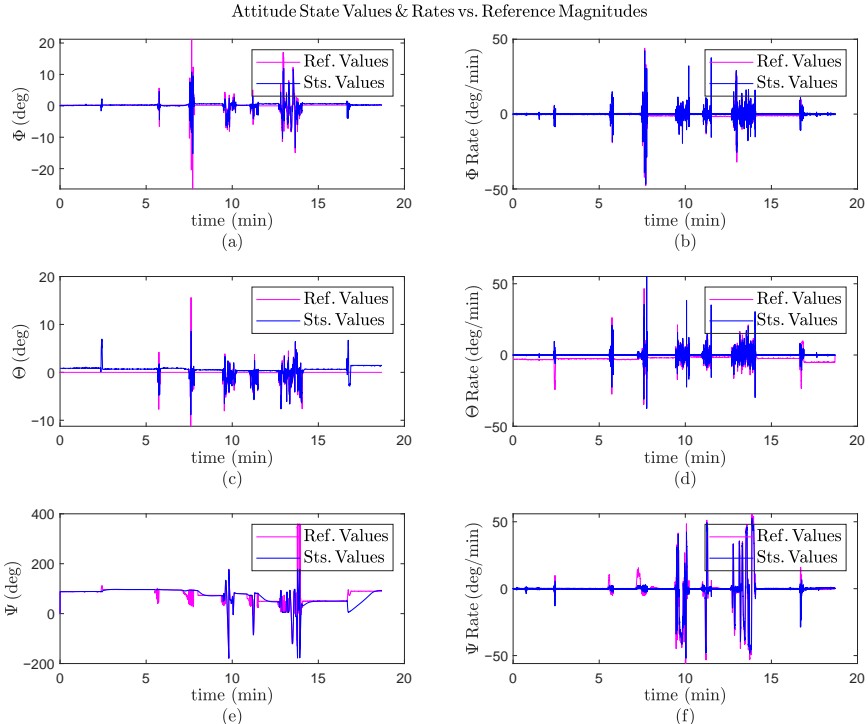

**Figure 15.** Practical results with intermediate disturbance, during 18 min; (**a**) the roll (*φ*) angle variation versus time; (**b**) the pitch (*θ*) angle variation versus time; (**c**) the yaw (*ψ*) angle variation versus time; (**d**) the roll rate (*φ̇*) variation versus time; (**e**) the pitch rate (*θ̇*) variation versus time; (**f**) the yaw rate (*ψ̇*) variation versus time.

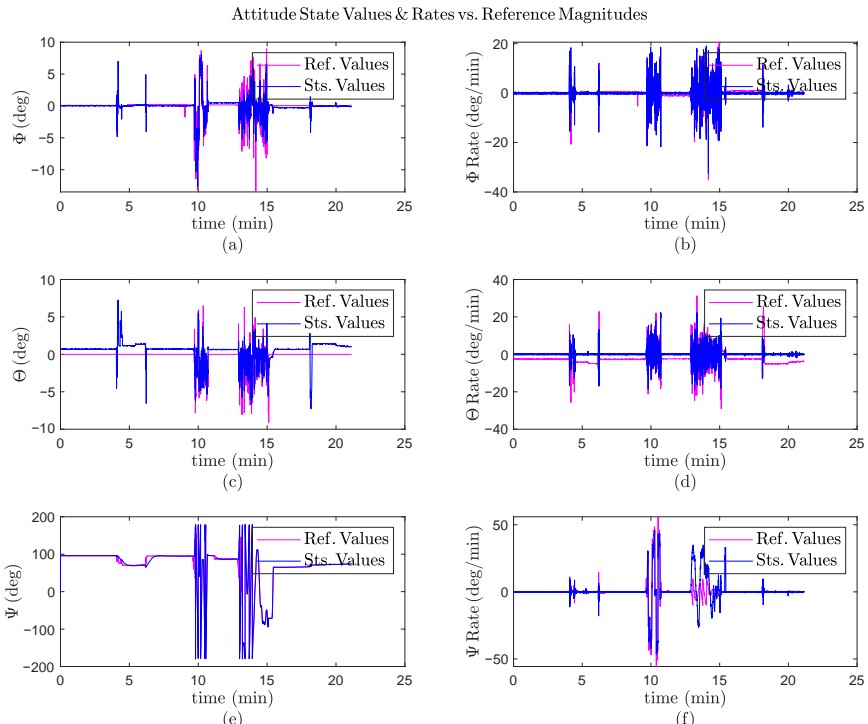

**Figure 16.** Practical results with an EDF failure, during 22 min; (**a**) the roll (*φ*) angle variation versus time; (**b**) the pitch (*θ*) angle variation versus time; (**c**) the yaw (*ψ*) angle variation versus time; (**d**) the roll rate (*φ̇*) variation versus time; (**e**) the pitch rate (*θ̇*) variation versus time; (**f**) the yaw rate (*ψ̇*) variation versus time.



## 8. Discussion and Conclusions

This paper concentrated on the attitude control system of a medium-scale hexacopter powered by EDFs employing an MRAC algorithm with proven asymptotic stability by introducing an appropriate Lyapunov candidate. During the investigation, complete aerodynamic modeling was performed to choose suitable ducted fans and prevent the fluid injection system from making the UAV unstable. Furthermore, the controller's performance was examined through the Gazebo dynamic space simulation, estimating the controller's initial parameters and maintaining the safety of the practical tests. According to the results obtained from simulation and practice, the attitude regulator functions are adequately acceptable, with errors of less than 5% in most cases. Additionally, an EDF failure was applied to the system in both spaces, but the controller quickly brought everything back to stability. This research revealed the possibility of controlling a rectangular hexacopter when powerful engines are mounted to the system, and even if one of the motors fails, the flight can be recovered.

**Author Contributions:** Conceptualization, M.S.A.I., M.A.L., M.M.A.E.K., R.K., A.R.R., P.C., P.F.P. and M.M.; data curation, M.S.A.I. and M.A.L.; formal analysis, M.S.A.I., M.A.L., A.R.R. and M.M.A.E.K.; funding acquisition, P.F.; investigation, M.S.A.I., M.A.L., A.R.R., P.C., P.F.P. and M.M.; methodology, M.S.A.I., M.A.L., A.R.R., M.M.A.A, R.K., P.C. and M.M.; project administration, M.M.A.E.K., R.K., P.C., P.F.P. and M.M.; resources, P.C. and P.F.; software, M.S.A.I. and M.A.L.; supervision, A.R.R., P.C., P.F.P. and M.M.; validation, M.S.A.I., M.A.L., A.R.R., M.M.A.E.K., R.K., P.C., P.F.P. and M.M.; visualization, M.S.A.I. and M.A.L.; writing—original draft, M.S.A.I., M.A.L. and A.R.R.; writing—review and editing, A.R.R., P.C., and M.M. All authors have read and agreed to the published version of the manuscript.

**Funding:** This research was supported by the European Commission-funded program FASTER, under H2020 Grant Agreement 833507 and partially funded by the Community of Madrid through the project REF: IND2020/IND-17478 "Ayudas para la realización de Doctorados Industriales", specially regarding the work of the second and sixth authors. This work has also been supported by the project COMCISE RTI2018-100847-B-C21, funded by the Spanish Ministry of Science, Innovation and Universities (MCIU/AEI/FEDER, UE).

**Institutional Review Board Statement:** Not applicable.

**Informed Consent Statement:** Not applicable.

**Data Availability Statement:** Not available.

**Acknowledgments:** We are so thankful to the Drone Hopper company for supporting the project, the Carlos III University for providing the hanger and academic space to investigate, Adrian Revuelta for designing the exact models and assemblies, Antonio Buendia Ruiz for sharing precious experiences regarding the ducted fan analysis and documentation, and also the Flow Engineering Company for a deep aerodynamic analysis of the EDFs.

**Conflicts of Interest:** The authors declare no conflict of interest.

## Abbreviations

The following abbreviations are used in this manuscript:

| | |
|---|---|
| AoA | Angle of attack. |
| ANCF | Absolute nodal coordinate formulation. |
| AP | Autopilot. |
| CoG | Center of gravity. |
| CoM | Center of mass. |
| ECEF | Earth centered Earth fixed frame. |
| EDF | Electric ducted fan. |
| EKF | Extended Kalman filter. |
| ESC | Electrical speed controller. |

| FAFSMC | Fuzzy adaptive fixed-time sliding mode controller. |
|--------|---------------------------------------------------|
| FE | Final element. |
| MRAC | Model reference adaptive controller. |
| NED | North east down. |
| SPR | Strictly positive real. |
| TSR | Tip speed ratio. |
| VTOL | Vertical takeoff and landing. |

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
