# Peer review of "Medium-Scale UAVs: A Practical Control System Considering Aerodynamics Analysis"

_drones, doi:10.3390/drones6090244_

Round 1
Reviewer 1 Report
This paper introduces a medium-scale hexacopter with its equations. As this work is a practical study and the results look fine, I suggest accepting it with a minor revision. I have some comments which I hope will be helpful to improve the paper:
1. As the main concept in the control system is Lyapunov-based, I would suggest writing some sentences in the introduction about other Lyapunov-based control methods, such as the following papers:
a) Rosales, Claudio, Carlos Miguel Soria, and Francisco G. Rossomando. "Identification and adaptive PID Control of a hexacopter UAV based on neural networks." International Journal of Adaptive control and signal processing 33, no. 1 (2019): 74-91.
b) Nguyen, Ngoc Phi, Nguyen Xuan Mung, Le Nhu Ngoc Thanh Ha, and Sung Kyung Hong. "Fault-Tolerant Control for Hexacopter UAV Using Adaptive Algorithm with Severe Faults." Aerospace 9, no. 6 (2022): 304.
c) Abadi, Ali Soltani Sharif, Pooyan Alinaghi Hosseinabadi, and Saad Mekhilef. "Fuzzy adaptive fixed-time sliding mode control with state observer for a class of high-order mismatched uncertain systems." International Journal of Control, Automation and Systems 18, no. 10 (2020): 2492-2508.
2. The curve of the control inputs in the simulation results should be added.
3. The discussion section can be improved.
Author Response
Reviewer 1:
This paper introduces a medium-scale hexacopter with its equations. As this work is a practical study and the results look fine, I suggest accepting it with a minor revision. I have some comments which I hope will be helpful in improving the paper:
Reviewer point #1:
As the main concept in the control system is Lyapunov-based, I would suggest writing some sentences in the introduction about other Lyapunov-based control methods, such as the following papers:
- a) Rosales, Claudio, Carlos Miguel Soria, and Francisco G. Rossomando. "Identification and adaptive PID Control of a hexacopter UAV based on neural networks." International Journal of Adaptive control and signal processing 33, no. 1 (2019): 74-91.
- b) Nguyen, Ngoc Phi, Nguyen Xuan Mung, Le Nhu Ngoc Thanh Ha, and Sung Kyung Hong. "Fault-Tolerant Control for Hexacopter UAV Using Adaptive Algorithm with Severe Faults." Aerospace 9, no. 6 (2022): 304.
- c) Abadi, Ali Soltani Sharif, Pooyan Alinaghi Hosseinabadi, and Saad Mekhilef. "Fuzzy adaptive fixed-time sliding mode control with state observer for a class of high-order mismatched uncertain systems." International Journal of Control, Automation and Systems 18, no. 10 (2020): 2492-2508.
Author response #1:
The introduction part has been improved, as shown in the file enclosed.
Reviewer point #2: The curve of the control inputs in the simulation results should be added.
Author response #2: The reference values highlighted with PURPLE color are the controller inputs. The context is improved to lead the reader to analyze the results, as shown in the file enclosed.
Reviewer point #3: The discussion section can be improved.
Author response #3: The conclusion part has been improved, as shown in the file enclosed.
Reviewer 2 Report
Introduction
- firstly, refers to the UAVs classification, and the necessity of power for their mission, in direct connection with the studies from the literature;
- then, the problem of the energy consumption (thermal engines, electrical engines, ducted fan etc.) is approached, with advantages and disadvantages;
- the purpose of the paper, including the organization in sections, are well formulated.
Medium scale multicopters
- attention to the title of the section (capitalize first letter);
- the components of the Fan Hopper are presented;
- the aerodynamics considerations are well developed; here, the Fig. 3a is not very clear (maybe a zoom would be helpful).
Hexacopter Time Varying Dynamics
- here, is not very clear which part of the text refers to the Fig. 5; usually, all diagrams are mentioned in the text of the paper, followed by the diagram herself and description;
- in equation (2), attention to the description of the notation used for the density (with/without subscript).
The Proposed Control Strategy
- the diagram of adaptive controller, the space state equation and the control rule are well described;
- in terms of stability, the Lyapunov mechanism is also, well applicated.
Simulation Results
- this section describes the simulation mechanism, based on Gazebo;
- the scenarios and the simulation results (Fig. 8) are well chosen.
Hardware Design
- all technical details (angles, torques, CAD model, 3D views) are well chosen and are helpful to understand/explain the results;
- the diagrams are inserted in order but the reference to them are as follow: Fig. 8, Fig. 10, Fig. 12, then Fig. 9 (this is just a problem of organization).
Practical Results
- the practical results obtained after multiple tests without/with disturbance (Fig. 14, 15) are compelling;
- maybe, with such amount of data, the tuning values of the PID could complete this section.
The References are suggestive for the topic. Random check for: 1. Fahlstrom, P.G.; 4. DeSmidt, H.; 8. Luna, M.A.; 12. Chen, C.; 19. Bresciani, T.; 22. Wang, H.; 25. Andrievsky, B.R.
Author Response
Point 1:
Introduction
- firstly, refers to the UAVs classification, and the necessity of power for their mission, in direct connection with the studies from the literature;
- then, the problem of the energy consumption (thermal engines, electrical engines, ducted fan etc.) is approached, with advantages and disadvantages;
- the purpose of the paper, including the organization in sections, are well formulated.
Medium scale multicopters
- attention to the title of the section (capitalize first letter);
- the components of the Fan Hopper are presented;
- the aerodynamics considerations are well developed; here, the Fig. 3a is not very clear (maybe a zoom would be helpful).
Response 1: The points are improved, as shown in the file enclosed, and in Figure 3(a), another view is added, but the quality of all figures is suitable for zooming to see all the details.
Point 2:
Hexacopter Time Varying Dynamics
- here, is not very clear which part of the text refers to the Fig. 5; usually, all diagrams are mentioned in the text of the paper, followed by the diagram herself and description;
- in equation (2), attention to the description of the notation used for the density (with/without subscript).
Response 2: Figure 5 was described above, and now the description is improved, as shown in the file enclosed.
Also, the subscript of the density is corrected, as shown in the file enclosed.
Point 3:
The Proposed Control Strategy
- the diagram of adaptive controller, the space state equation and the control rule are well described;
- in terms of stability, the Lyapunov mechanism is also, well applicated.
Simulation Results
- this section describes the simulation mechanism, based on Gazebo;
- the scenarios and the simulation results (Fig. 8) are well chosen.
Hardware Design
- all technical details (angles, torques, CAD model, 3D views) are well chosen and are helpful to understand/explain the results;
- the diagrams are inserted in order but the reference to them are as follow: Fig. 8, Fig. 10, Fig. 12, then Fig. 9 (this is just a problem of organization).
Practical Results
- the practical results obtained after multiple tests without/with disturbance (Fig. 14, 15) are compelling;
- maybe, with such amount of data, the tuning values of the PID could complete this section.
The References are suggestive for the topic. Random check for: 1. Fahlstrom, P.G.; 4. DeSmidt, H.; 8. Luna, M.A.; 12. Chen, C.; 19. Bresciani, T.; 22. Wang, H.; 25. Andrievsky, B.R.
Response 3: About the tuning of PID coefficients, all of them are several times tuned as mentioned in the file enclosed, it is noted in the paper.
Then the figures are reordered following the description, as shown in the file enclosed.
